# The Driving Influence of Multi-Dimensional Urbanization on PM_2.5_ Concentrations in Africa: New Evidence from Multi-Source Remote Sensing Data, 2000–2018

**DOI:** 10.3390/ijerph18179389

**Published:** 2021-09-06

**Authors:** Guoen Wei, Pingjun Sun, Shengnan Jiang, Yang Shen, Binglin Liu, Zhenke Zhang, Xiao Ouyang

**Affiliations:** 1College of Geography and Ocean Sciences, Nanjing University, Nanjing 210023, China; dg1927034@smail.nju.edu.cn (G.W.); dz1627002@smail.nju.edu.cn (S.J.); y_shen@yeah.net (Y.S.); DG1827018@smail.nju.edu.cn (B.L.); 2Institute of African Studies, Nanjing University, Nanjing 210023, China; 3College of Geographical Sciences, Southwest University, Chongqing 400700, China; sunpj031@nenu.edu.cn; 4International Institute for Earth System Science, Nanjing University, Nanjing 210023, China; 5Hunan Institute of Economic Geography, Hunan University of Finance and Economics, Changsha 410205, China

**Keywords:** PM_2.5_ concentrations, multi-dimensional urbanization, spatial regression model, spillover effect, African region

## Abstract

Africa’s PM_2.5_ pollution has become a security hazard, but the understanding of the varying effects of urbanization on driven mechanisms of PM_2.5_ concentrations under the rapid urbanization remains largely insufficient. Compared with the direct impact, the spillover effect of urbanization on PM_2.5_ concentrations in adjacent regions was underestimated. Urbanization is highly multi-dimensional phenomenon and previous studies have rarely distinguished the different driving influence and interactions of multi-dimensional urbanization on PM_2.5_ concentrations in Africa. This study combined grid and administrative units to explore the spatio-temporal change, spatial dependence patterns, and evolution trend of PM_2.5_ concentrations and multi-dimensional urbanization in Africa. The differential influence and interaction effects of multi-dimensional urbanization on PM_2.5_ concentrations under Africa’s rapid urbanization was further analyzed. The results show that the positive spatial dependence of PM_2.5_ concentrations gradually increased over the study period 2000–2018. The areas with PM_2.5_ concentrations exceeding 35 μg/m^3^ increased by 2.2%, and 36.78% of the African continent had an increasing trend in Theil–Sen index. Urbanization was found to be the main driving factor causing PM_2.5_ concentrations changes, and economic urbanization had a stronger influence on air quality than land urbanization or population urbanization. Compared with the direct effect, the spillover effect of urbanization on PM_2.5_ concentrations in two adjacent regions was stronger, particularly in terms of economic urbanization. The spatial distribution of PM_2.5_ concentrations resulted from the interaction of multi-dimensional urbanization. The interaction of urbanization of any two different dimensions exhibited a nonlinear enhancement effect on PM_2.5_ concentrations. Given the differential impact of multi-dimensional urbanization on PM_2.5_ concentrations inside and outside the region, this research provides support for the cross-regional joint control strategies of air pollution in Africa. The findings also indicate that PM_2.5_ pollution control should not only focus on urban economic development strategies but should be an optimized integration of multiple mitigation strategies, such as improving residents’ lifestyles, optimizing land spatial structure, and upgrading the industrial structure.

## 1. Introduction

Since the turn of the century, global value chains and production networks have accelerated their penetration into various regions. Globalization has entered a new era of inclusive development, propelling urbanization at tremendous vitality levels [1,2]. Waves of rapid urbanization have been experienced in regions, including East Asia, South Asia, Southeast Asia, and Central America [3]. With urban spaces increasingly expanding into the suburbs, huge populations have migrated to urban built-up areas, which has considerably improved residential living conditions [4]. However, in the past few decades, increased urbanization activities, such as agriculture economy, industrial production, residential life, commercial trade, and energy emissions, have been proved to be important factors to accelerate global PM_2.5_ levels [5,6,7,8]. Air pollution has become the fifth leading cause of death in the world. Long-term exposure to PM_2.5_ causes more than 4.2 million premature deaths annually, with associated diseases such as ischemic heart disease, cerebrovascular disease, and lower respiratory tract infection [9,10]. Given the association of urbanization and PM_2.5_ levels, policymakers have to be more aware and responsive to address the needs of human well-being and air quality sustainability.

Previous studies have demonstrated that urbanization can affect PM_2.5_ concentrations, resulting different degrees of response in air pollution control for various regions, such as China’s “Air Pollution Prevention and Control Action Plan” issued in 2013 and the US “Clean Air Act” policies [11,12]. Some studies have explored the driving impact of urbanization on PM_2.5_ pollution using a single dimension, such as human activity intensity (i.e., population urbanization) and urban land expansion intensity (i.e., land urbanization) [13]. Few treated urbanization as a comprehensive system comprising economic growth, population agglomeration, urban land expansion, and other elements, combined with natural factors to analyze the driving mechanism of PM_2.5_ concentrations, which led to the limited comprehensiveness of the mechanism analysis [14,15].

In recent years, remote sensing data (e.g., PM_2.5_ satellite inversion data, nighttime lighting data, land use data, and impervious surface data) in the interaction research between urbanization and PM_2.5_ concentrations has become widely used. Compared to traditional statistical data of administrative regions, this has greatly improved the objectivity and comparability in spatial distribution research [16]. However, in most instances, spatial distribution analyses are conducted at the country or city level, which weakens the refinement degree of spatial expression of remote sensing data and resulted a great compromise of data analysis [17,18]. Studies rarely utilize joint analysis on administrative units and grid units, resulting in poor integration of decision-making reference value and refined expressions.

Spatial regression models, such as the spatial lag model (SLM), spatial error model (SEM), and spatial Durbin model (SDM), have been used to measure the impact of spatial differentiation of urbanization on PM_2.5_ concentrations [18,19]. However, the spillover effect of urbanization on PM_2.5_ concentrations in adjacent regions has often been overlooked, as well as the different influencing forces and interaction effects of multi-dimensional urbanization. Such oversight has made it difficult to fully understand the endogenous and exogenous mechanisms of the urbanization effect on PM_2.5_ concentrations.

International investors heavily favor Africa’s potential due to its enormous labor force, raw materials, and markets. Establishing a free trade zone (FTZ) and supporting the development of small and medium-sized enterprises has become the consensus of many African countries to stimulate national economic growth. Urban construction and infrastructure development are in full swing in Johannesburg, Cairo, Nairobi, and Algiers, resulting in significant improvements to people’s living conditions and general welfare [20]. However, increased urbanization has also resulted in higher levels of air pollution. Exhaust emissions from old cars, open garbage incineration, coal smoke pollution, and industrial pollutant emissions have clouded the “ecological Mainland”, making air pollution one of the most serious social hidden dangers in the continent, along with AIDS, malaria, poverty, and war [21,22,23]. The backwardness in environmental technology and the low investments in environmental protection have aggravated the negative impact of Africa’s PM_2.5_ pollution, estimated to cost about USD 450 billion in annual economic losses [24]. The issue of PM_2.5_ pollution has received increasing attention from different international organizations, and various governments have put forth their ambitions for ecologically sustainable development on the “African Agenda 2063” [25]. However, social awareness and attention remain insufficient to the driving effects of multi-dimensional urbanization on PM_2.5_ concentrations in Africa.

To sum up, previous studies have helped enhance the understanding of urbanization effects on PM_2.5_ concentrations and have supported urban policymakers in emerging developing countries towards human–land harmony and sustainable development. However, in the context of rapid urbanization, the discussion on Africa’s PM_2.5_ pollution is far from being settled due to the following reasons: (1) the measurement of urbanization remains limited and disparate. Significant improvements are required to comprehensively analyze the impact of urbanization on PM_2.5_ concentrations in Africa from a broader perspective (including population, land, and economy); (2) the joint analysis of grid and administrative units has not received much attention. Such analysis can achieve the integration of decision-making reference value and refined expression; and (3), few studies have integrated multiple spatial regression models to study the difference between the direct and spillover effects of urbanization on PM_2.5_ concentrations and analyzed the interaction effects between multi-dimensional urbanization, although this can provide support for the cross-regional joint control of air pollution in Africa.

To address these knowledge gaps, this study explores the spatial distribution characteristics of multi-dimensional urbanization (population urbanization, land urbanization, economic urbanization) and PM_2.5_ concentrations in Africa for 2000–2018 using grid and administrative units. Along with the impact of natural factors (e.g., altitude, elevation, rainfall, vegetation coverage), this study analyzes the effects of urbanization on PM_2.5_ concentrations using spatial regression models and investigates the differential impact and interaction of multi-dimensional urbanization on PM_2.5_ concentrations. The main objectives of this study are as follows: (1) to characterize the spatio-temporal pattern and spatial dependence features of PM_2.5_ concentrations and multi-dimensional urbanization; (2) to explore the driving mechanism of PM_2.5_ concentrations under the rapid modernization in Africa and determine whether the effects of urbanization on PM_2.5_ concentrations would spillover among neighboring areas; and (3) to compare the impact and interaction of urbanization in different dimensions on PM_2.5_ concentrations. The results of this study can provide reference for African countries to optimize the coordination relationship between urbanization and air quality systems.

## 2. Materials and Methods

### 2.1. Materials

Table 1 summarizes the variables used in this study. The datasets were obtained from the following sources:
(1)PM_2.5_ concentrations. The Atmospheric Composition Analysis Group (ACAG) provided the PM_2.5_ concentrations (V4.GL.03, 0.05° × 0.05°, Contains “all ingredients”) from the African satellite corrected by geographically weighted regression (GWR) for 1998–2018 (http://fizz.phys.dal.ca/~atmos/martin/?page_id=175) (accessed on 15 December 2020) [26,27]. The mean annual PM_2.5_ concentration grid data were obtained by vector clipping and calculated using Python 2.7 (http://www.python.org) (accessed on 20 December 2020).(2)Urbanization. Based on the availability of remote sensing data and existing research, multiple dimensions of urbanization were measured using various indicators, including population urbanization, land urbanization, and economic urbanization [28]. The population urbanization level was depicted by the population density, which was considered to be the most direct indicator of the spatial pattern of population distribution [29]. Its data were obtained from the LandScan Global Population Project, developed by the Department of Energy Oak Ridge National Laboratory (ORNL) in Tennessee, USA based on a combination of geographic information systems, image analysis, and multivariate zoning density models at 1 km spatial resolution (https://www.satpalda.com/product/landscan/) (accessed on 25 December 2020) [30]. The land urbanization level, characterized by the degree of the artificial impervious surface coverage (impervious surface coverage = impervious surface area/total area) [31], was calculated using a 30 m high-resolution artificial impervious surface product by Professor Gong Peng [32]. This product was produced by long-time Landsat optical images (nearly 1.5 million scenes) and other auxiliary data (http://data.ess.tsinghua.edu.cn/) (accessed on 25 December 2020) and has been used in various studies to quantify the expansion intensity of urban areas [33]. For economic urbanization, several studies have shown the feasibility of analyzing regional economic levels using nighttime light intensity [34,35]. Since DMSP/OLS (2000–2013a) and NPP/VIIRS (2013–2018a) are different datasets, continuity correction is required. The global nighttime light dataset (1992–2018a) by Li et al. (2020), accessed from the Scientific Data platform (Nature Group), was used as the data source [36]. In this dataset, a sigmoid function was used to establish the continuity relationship between the DMSP and VIIRS datasets after noise reduction. Given a consistent spatial resolution and radiation characteristics, the performance evaluation showed that the dataset is reliable and can be used stably for long-term global research (https://www.nature.com/articles/s41597-020-0510-y) (accessed on 25 December 2020).(3)Natural indicators. In addition to urbanization, the agglomeration and diffusion of PM_2.5_ concentrations have been shown to be highly related to natural factors such as topography, meteorological factors, and vegetation coverage. Xu et al. (2018) concluded that the radiative cooling effect of aerosols caused by low temperatures in winter promotes the accumulation of PM_2.5_ concentrations [37]. Fang et al. (2020) found that increased forest coverage can reduce PM_2.5_ concentrations [17]. Zhou et al. (2016) proved that slope and altitude play an important role in blocking PM_2.5_ pollutants [38]. To reduce omitted variable bias, natural indicators, including normalized differential vegetation index (*ndvi*), cumulative precipitation (*pre*), and elevation (*ele*), were used as control variables in the spatial regression model. As natural control factors, their statistically negative or positive impact would not greatly influence the relationship between urbanization and PM_2.5_ concentrations. The *ndvi* data were derived from MODIS monthly NDVI product (MOD13A3-6) at 1 km spatial resolution (https://ladsweb.modaps.eosdis.nasa.gov/search/order/1/MOD13A3-6) (accessed on 5 December 2020). The cumulative precipitation data were obtained from the National Earth System Science Data Center (CN) (http://www.geodata.cn/index.html) (accessed on 5 December 2020), with a spatial resolution of 2.5 points. The elevation and slope data were derived from the GDEMV2 DEM digital elevation product of the Computer Network Information Center of the Chinese Academy of Sciences (http://www.gscloud.cn) (accessed on 5 December 2020) at 30 m spatial resolution and were processed using ArcGIS.


### 2.2. Methods

The research structure consists of three parts: (1) Theil–Sen median trend degree and spatial autocorrelation methods were used to analyze the spatio-temporal evolution and spatial dependence patterns of PM_2.5_ concentrations and the multi-dimensional urbanization in Africa; (2) spatial regression models (SEM, SLM, and SDM) were used to reveal the driving mechanism of PM_2.5_ concentrations under the rapid modernization; and (3) joint spatial regression model and geographic detector were employed to analyze the differential impact and interaction effects of multi-dimensional urbanization on PM_2.5_ concentrations.

#### 2.2.1. Theil–Sen Median Trend Degree

The spatio-temporal evolution trend (SET) of PM_2.5_ concentrations and multi-dimensional urbanization were analyzed at the grid level. The Theil–Sen median trend degree is a robust non-parametric statistical trend calculation method that has a strong resistance to calculation error [39]. The Theil–Sen median trend degree was calculated as follows:(1)$$S_{PM2.5}=Median\left(\frac{x_{i}-x_{j}}{i-j}\right),1998\leq j<i\leq 2018$$
where *x_i_* is the observations at grid level in *i* year; and *S* is the median of the slopes of *n*(*n* − 1)/2 data combinations. *S*-values significantly above zero indicate substantial increases in grid index, while *S*-values significantly below zero suggest substantial declines in grid index over the period 2000–2018.

#### 2.2.2. Spatial Autocorrelation Methods

To explore the spatial dependence pattern (SDP) of PM_2.5_ concentrations, the global autocorrelation Moran’s *I* and hot/cold spots method were determined at the administrative unit level to characterize the global and local spatial agglomeration [40]. Global autocorrelation Moran’s *I* [−1,1] was calculated as follows:(2)$$I=\frac{n\sum_{i=1}^{n}\sum_{j\neq 1}^{n}W_{ij}(x_{i}-\bar{x})(x_{j}-\bar{x})}{\sum_{i=1}^{n}\sum_{j=1}^{n}W_{ij}\sum_{i=1}^{n}{\left(x_{i}-x\right)}^{2}}$$
where *n* is the number of grid units; *W_ij_* is spatial weight matrix; *x_i_* and *x_j_* are the observations of PM_2.5_ concentrations at the grid level. Moran’s *I* significantly above zero indicates positive spatial dependence, significantly below zero indicates negative spatial dependence, and zero means no agglomeration feature.

Hot/cold spot analysis is performed using the Getis-OrdGi* tool, which characterizes the grids into four types: hot spot, cold spot, sub-hot spot, and sub-cold spot [41]. The Getis-OrdG_i_* (*G_i_**(*d*)) was calculated using:(3)$${G_{i}}^{*}(d)=\sum_{i=1}^{n}W_{ij}(d)x_{i}/\sum_{i=1}^{n}x_{j}$$

Values above zero indicate hot spots while values below zero indicate cold spots. Given the empirical selection from previous studies and the suitability comparison of measurement results from different methods, the spatial relations and spatial weights were constructed using the Fixed Distance and the Euclidean distance methods [40].

#### 2.2.3. Spatial Regression Analysis

Spatial regression analysis has an advantage over linear regression given its ability to consider the influence of spatial dependence on regression indexes. Based on the spatial autocorrelation characteristics of variables, this study introduced grid data into SEM, SLM, and SDM. The SEM focuses on the error of dependent variables in adjacent regions to explain PM_2.5_ pollution on the areas, while the SLM focuses on the spatial spillover effects of urbanization and natural factors in the PM_2.5_ concentrations of adjacent regions. For more details on the SLM and SEM; see Lou et al. (2016) [42]. As a general form of SLM and SEM, the SDM can integrate the endogenous and exogenous characteristics of variables and is given by the expression:(4)$$Y=\rho WY+X\beta +WX\theta +\alpha K_{n}+\varepsilon$$
where *Y* (dependent variable) is the PM_2.5_ concentrations of the grid level; *X* consists of key explanatory variables and natural control variables; *ρ*, *α*, *β*, and *θ* are the parameters to be estimated; *ε* is a normally distributed disturbance term with a diagonal covariance matrix; *W* is the spatial weight matrix reflecting the data spatial structure and spatial relationship among locations; *WY* is the spatial lag dependent variable; and, *WX* is the spatial lag independent variable. As a general form of SLM and SEM, the SDM can transform each other [43]. The likelihood ratio (LR) was used to examine the applicability of the models: (1) If *ρβ* + *θ* = 0 passes the LR significance test, SDM can be reduced to the SEM. (2) If *θ* = 0 passes the LR significance test, SDM can be reduced to the SLM. If both are rejected, the SDM will be more suitable for the regression analysis of this study [15].

The following procedures were undertaken to improve the rationality for spatial regression analysis: (1) the variance inflation factor (VIF) and Condition Index were calculated in the SPSS software to test multi-collinearity among variables. The results showed that the VIF of each variable was less than 7.5 and the Condition Index was less than 10; (2) normalization was performed for both dependent and independent variables to reduce the heteroscedasticity; and (3) analysis and selection of the regression model were performed based on Du et al. (2019) [15]. After passing the Lagrangian multiplier (LM) test and residual spatial autocorrelation test, the LR was used to test the significance of the hypothesis and determine whether the SDM can be reduced to SLM or SEM. Spatial regression models and related statistical tests were performed in MATLAB.

#### 2.2.4. Geographic Detector

The geographical detector consists of four modules: factor detector, risk detector, interaction detector, and ecological detector [44]. The interaction detector can determine the mode and direction of the variable interaction based on the quantitative relationship between the explanatory power of the bivariate interaction and the univariate explanatory power. The evaluation criteria of interaction mode are summarized in Table 2 [45]. The interaction detector was used to measure the interaction influence of population urbanization, land urbanization, economic urbanization on the PM_2.5_ concentrations in Africa, calculated using the formula:(5)$$P_{D}=1-\frac{\sum_{i=1}^{L}N_{h}\sigma_{h}^{2}}{N\sigma^{2}}$$
where *D* is the urbanization level in different dimensions, *P_D_* is the explanatory power intensity of urbanization factor, *h* is the number of grid data classifications based on natural breaks (Jenks), *N_h_* is the number of category *h*, *N* is the number of categories in all grid units, *σ_h_*^2^ is the variance of *Y* values for class h, and *σ*^2^ is the variance of *Y* values for the total grid. The operation was calculated in the GeoDetector.

### 2.3. Study Area

The study area is the entire African continent, composed of 55 countries. Using the geographic division standard created by the United Nations, the continent can be subdivided into five regions: Northern Africa, Western Africa, Southern Africa, Eastern Africa, and Central Africa (Table 3). For the union of administrative and grid levels, a 50 × 50 km grid layer was established using ArcGIS, resulting in 13,633 grid units for the entire continent.

## 3. Results

### 3.1. Spatio-Temporal Distribution and Spatial Dependence of PM_2.5_ Concentration and Urbanization under the Rapid Urbanization

Figure 1 shows the spatial distribution pattern of average PM_2.5_ concentrations for 2000–2018, which exhibits a strong geographical heterogeneity for PM_2.5_ concentrations over the 19 year study period. According to the Air Quality Guidelines by the World Health Organization (WHO), areas with PM_2.5_ concentrations of 35 μg/m^3^ (or more) have considerably higher mortality rates than those with less than 10 μg/m^3^. The PM_2.5_ concentrations in the Sahara Desert, the Gulf of Guinea, the Niger River Basin, and the Chad Basin far exceeded WHO’s recommended threshold limit of 35 μg/m^3^, while those in eastern and southern Africa were mostly below 10μg/m^3^. Figure 2 compares the classified areas of the mean annual PM_2.5_ concentrations for the different years. Areas with mean annual concentrations above 35 μg/m^3^ increased from 43.25% in 2000 to 44.21% in 2018, equivalent to an annual growth rate of 0.12%. Extremely polluted areas with average concentrations exceeding 75 μg/m^3^ were reduced by 3.32%, from 5.98% in 2000 to 2.66% in 2018. In comparison, areas with concentrations ranging from 35 to 65 μg/m^3^ increased by 27.7% (29.31% in 2000 compared to 37.43%), equivalent to an annual growth rate of 1.35%.

The spatial distribution of PM_2.5_ concentrations at the national level (Figure 3) showed small cluster changes in PM_2.5_ concentrations (based on natural breaks classification), with significant drops of mean concentrations in the Democratic Republic of Congo, the Congo, and the Central African Republic. PM_2.5_ concentrations were low in Southern and Eastern African countries, while many Northern, Western, and Central African countries had mean PM_2.5_ concentrations greater than 35 μg/m^3^. In particular, Chad, Niger, and Mali around the Sahara Desert and Nigeria, Côte d’Ivoire, Ghana, and Togo along the Gulf of Guinea had average PM_2.5_ concentrations greater than 50 μg/m^3^.

The results also showed that the levels of population agglomeration, land expansion, and economic construction exhibited varying degrees of spatial heterogeneity. (1) Population agglomeration (Figure 4): from 2000 to 2018, the mean population agglomeration level in African countries increased by 44.1%. Among them, Eastern Africa had the highest population agglomeration average level and fastest growth rate, which the population density increased to 149 persons/km^2^ and a growth rate of 44.4% from 2000 to 2018. (2) Land expansion (Figure 5): from 2000 to 2018, the mean land expansion level among African countries increased by 54.7%. By 2018, the high value areas were concentrated in Northern and Southern Africa and in the coastal countries along the Gulf of Guinea. In 2018, the impervious surface coverage (ISC) in Northern Africa was 0.0031, and 0.0027 in the Southern African region, representing an increase of 47.5% and 35.08%, respectively. The countries with the highest ISC included Mauritius, Ghana, Tunisia, South Africa, Egypt, and Nigeria, with values exceeding 0.0053. (3) Economic construction (Figure 6): from 2000 to 2018, the mean economic construction level among African countries increased by 359%, of which Central and Western Africa increased the most, with 1051.9% and 314.4%, respectively. High-value clusters formed gradually across the Central Region, from Uganda to Senegal. Other high-value areas can also be found in Northern Africa (Tunisia, Morocco, Egypt) and Southern Africa (South Africa, Lesotho, Swaziland).

Theil–Sen trend analysis showed the Sen trend index of Africa’s PM_2.5_ concentrations levels had been dominated by a growth trend over the 19 year study period, with 36.78% of the region experiencing positive growth and 6% of which having rapid growth (Figure 7). Ethiopia and parts of the Congo River Basin (Chad, Republic of Congo, the Democratic Republic of the Congo) registered massive increasing PM_2.5_ pollution, while those along the Mediterranean coast and the Gulf of Guinea (Togo, Ghana, Benin, Nigeria) had gradually declined. In terms of the population urbanization level, 46.32% of the continent posted a positive change. Rapid growths were concentrated in Nigeria, Central Ethiopia, parts of the African Great Lakes region (Rwanda, Burundi, Uganda, and Kenya), and international metropolises, such as Johannesburg, Cairo, and Rabat. The continent had relative stability in terms of land urbanization level, while 32.11% posted positive changes in economic urbanization level. Both land urbanization and economic urbanization development were found to have significant coastal–inland distribution heterogeneity, while growth areas were mainly found in the Mediterranean coast (a coastal city belt composed of Rabat, Algiers, Tripoli, Tunisia, and Cairo), Southern Africa (South Africa, Sri Lanka, Lesotho), and the Gulf Coast of Guinea.

To characterize the SDP of PM_2.5_ concentrations at the administrative level, we calculated Moran’s *I* for PM_2.5_ concentrations per province (Figure 8). The estimated Moran’s *I* values for 2000 and 2018 were 0.643 and 0.646, respectively (*p* < 0.001), indicating increased spatial dependence of PM_2.5_ concentrations. Spatial agglomeration characteristics of cold and hot spots were significant, with hot spots distributed in the west and cold spots in the east. A “crescent-shaped” dividing line between the hot spot and cold spot regions can be seen from Morocco’s Doukkala-Abda to Angola’s Bengo. For the given study period, the majority of the continent was categorized as cold spots, but its overall share has fallen by 6.1%, and the sub-cold spot area (95% confidence level) fell by 28.6%. By 2018, some areas along the dividing line shifted to sub-hot spots (95% confidence level), and the total hot spot area increased by 0.4%.

### 3.2. Driving Mechanism of PM_2.5_ Concentration under the Rapid Urbanization

The spatio-temporal distribution pattern confirmed the relationship between PM_2.5_ concentrations and urbanization in Africa. Most countries in Northern and Western Africa (especially along the Sahara Desert and the Gulf of Guinea) had PM_2.5_ concentrations well above 35 μg/m^3^ and registered high land expansion and economic construction levels. Hence, several questions arise as follows: (1) does urbanization impact PM_2.5_ concentrations in Africa? (2) How does this impact change during the study period? To answer the above questions, spatial regression models were constructed to further analyze the impact of urbanization on PM_2.5_ concentrations at the grid unit level by using urbanization as the key variable and natural factors as control variables. Three temporal cross-sections (i.e., 2000, 2010, 2018) were analyzed considering the data availability. Furthermore, referring to the study of Du et al. (2019), an indicator for comprehensive urbanization was established based on the multi-dimensional urbanization index (each dimension of the urbanization index was standardized, the former results were added, and the results were standardized again). The resulting value was used as the key explanatory variable (lncmpu) to reflect the impact of comprehensive urbanization on PM_2.5_ concentrations.

The results of the Lagrangian multiplier test (LM) and the robust Lagrangian multiplier test (Robust LM) in OLS estimates all rejected the null hypothesis that was no spatial lag term and no spatial error term at the significant level of 1%. The residual spatial autocorrelation test shows that the mean value of residual Moran’ I of OLS is 0.938 (at 1% confidence level), while the mean value of residual Moran’s I of spatial regression is closer to 0, which indicates the rapid decline of the spatial autocorrelation of the residuals, spatial regression models have to be introduced. The likelihood ratio (LR) for the SLM and SLM (namely, LR-SLM, LR-SEM) were both statistically significant (Table 4). This means that the SDM cannot be reduced to SLM or SEM and that SDM was more suitable in analyzing the driving mechanism analysis since it can simultaneously characterize the endogenous and exogenous interaction effects. The SDM calculation results (Table 4) for 2000, 2010, and 2018 show that the regression coefficient of comprehensive urbanization (lncmpu) is significantly greater than zero. In the periods 2000–2010 and 2010–2018, the comprehensive urbanization coefficient (lncmpu) grew by 5.88% and 79.63%, respectively. This means that urbanization had an increasing effect on PM_2.5_ concentrations and that the impact had intensified rapidly. Furthermore, as a control variable, the vegetation cover played an important role in mitigating PM_2.5_ pollution in Africa. From 2000 to 2018, the absolute value of vegetation coefficient (lnndvi) increased by 86.36%. The altitude and slope were also important factors mitigating the increase in PM_2.5_ concentrations, and their coefficients for 2018 were −0.034 (lnele) and −0.06 (lnele), respectively. The influence of cumulative precipitation on PM_2.5_ concentrations was weak, with coefficients (lnpre) for 2000 and 2018 being −0.017 and 0.015, respectively.

### 3.3. Differential Influence and Interaction of Multi-Dimensional Urbanization on PM_2.5_ Concentrations

After estimating the impact of comprehensive urbanization index on PM_2.5_ concentrations, other questions arose: (1) does each dimension of urbanization have same impact intensity on PM_2.5_ concentrations? (2) Does multi-dimensional urbanization have an interaction effect on PM_2.5_ concentrations? To assess the impact of the different urbanization dimensions on PM_2.5_ concentrations, spatial regression model and interaction detector were used with multi-dimensional urbanization as the explanatory variables and PM_2.5_ concentrations as the dependent variable. Based on SDM, Table 5 shows the different effects (i.e., direct effect and spillover effect) of various urbanization dimensions on PM_2.5_ concentrations for 2000, 2010, and 2018. The influence of multi-dimensional urbanization on PM_2.5_ concentrations typically ranked as follows: economic urbanization > land urbanization > population urbanization. This means that economic activity and urban land expansion had relatively stronger contributions to the PM_2.5_ concentrations in Africa.

For population urbanization, neither the direct effect (DEU) nor the spillover (SEU) effect of lnpd was significant in any period, and the absolute value of the elasticity coefficient for DEU approached zero (Table 5, row 2). For land urbanization, under statistical significance, the elasticity coefficient of the DEU increased from 0.015 in 2000 to 0.026 in 2018, equivalent to a 73.3% increase (Table 5, row 3). The elasticity coefficient of the SEU in 2000 was −1.015 (significant at the 1% level) but was not significant in 2010 and 2018. For economic urbanization, the DEH and SEH of lnntl increased over the years and were significant at the 1% level. From 2000 to 2018, the DEU and SEU for economic urbanization largely ranked first in multi-dimensional urbanization, and the elasticity coefficient increased by 35% and 98.5%, respectively (Table 5, row 4). Furthermore, the SEH had a stronger influence than the DEU in multi-dimensional urbanization, indicating that the impact of urbanization on PM_2.5_ concentrations was stronger on the adjacent areas than on the given region. For example, the elasticity coefficient of the DEU for population urbanization was 0.020 (significant at the 1% level), which was only 1/62 of the SEU.

Using data from 13,633 grid units, we employed an interaction detector to explore the interaction effects of multi-dimensional urbanization on PM_2.5_ concentrations. The results show that the interaction of any two different urbanization dimensions has a nonlinear enhancement effect on PM_2.5_ concentrations (Table 6). For example, the interaction’s explanatory power (A∩B) between PU and LU, PU and EU, and LU and EU in 2018 were 0.09, 0.63, and 0.46, while the combined explanatory power (A + B) were 0.022, 0.425, and 0.437, respectively. Moreover, the interaction effects of multi-dimensional urbanization were significantly different; the overall growth rate and the explanatory strength of the interaction effect in 2018 all can be ranked as follows: PU∩EU > LU∩EU > PU∩LU. The results suggest that the interaction effects between population agglomeration and economic construction and between urban land expansion and economic construction were important interaction mechanisms affecting the spatial distribution patterns of PM_2.5_ concentrations in Africa.

## 4. Discussion

### 4.1. Explanation for the Different Impact and Interaction Effect of Multi-Dimensional Urbanization on PM_2.5_ Concentrations

Why do the multiple urbanization dimensions show varying effects on PM_2.5_ concentrations in Africa? For population urbanization, the non-significant impact of DEU and SEU can be partially attributed to the inconsistent distribution pattern between PM_2.5_ concentrations and population urbanization. As shown in Figure 3 and Figure 4, many countries with high population density do not necessarily have high PM_2.5_ concentrations. In Eastern Africa, which has the highest average level and growth rate of population urbanization, and in high-value countries, such as Morocco, Tunisia, and Egypt, PM_2.5_ concentrations were less than 35 μg/m^3^. Meanwhile, some areas with high PM_2.5_ concentrations, including those around the Sahara Desert, such as Chad, Niger, and Mali, did not form high population agglomeration areas.

Furthermore, compared to economic urbanization and land urbanization, the spatial gravity center for population urbanization was farther from the PM_2.5_ concentrations center (see Figure 9). For 2018, the spatial distance of gravity centers between population urbanization and PM_2.5_ concentrations was 1240.3 km, which was 141.8% of the distance between economic urbanization and PM_2.5_ concentration gravity centers. This interesting finding could be attributed to the spatial difference in population growth and regional industrial development within Africa. In recent years, Kenya, Rwanda, Ethiopia, and other Eastern African countries have launched long-term development strategies suitable for the national economic conditions. These strategies focus on social livelihood, medical security, and employment in urban areas, resulting in considerable population shifts towards urban areas and the rapidly development of population urbanization [46]. According to the *African Statistical Yearbook (2019)*, Ethiopia and Kenya were major urban population growth areas in Africa in 2018, increasing by 12.9 million and 7.6 million since 2000. Meanwhile, the export economy rapidly developed among many Eastern African countries, where special economic zones, such as Free Trade Areas (FTA) and Export Processing Zones (EPZ), have been expeditiously created. For example, Ethiopia’s Eastern Industrial Park and Kenya’s Athi River Export Processing Zone were the typical manufacturing development model in Africa. Rapid construction of industrial areas and preferential tax policies in many Eastern African countries have helped to attract foreign investment, developing their service industries in tourism, logistics, and business services [47]. These economic actions have a smaller negative impact on air quality than extensive industrial production, causing the separation between Africa’s population growth center and PM_2.5_ pollution center.

The DEU and SEU of economic urbanization ranked first among the various urbanization dimensions. The DEU results suggest that PM_2.5_ concentrations were greatly affected by regional economic construction, which could be partly related to the negative environmental effects (e.g., waste gas pollution, energy consumption, smoke dust emission) caused by regional economic construction [49]. The Environmental Kuznets Curve (EKC) posits that environmental quality deteriorates with economic growth during the large-scale industrialization era. In the post-industrial era, negative scale effects would then be surpassed by technical and structural changes, and environmental quality gradually improves with economic growth. Most African regions are far from reaching the EKC inflection point, and the negative environmental effects brought by economic development are significant. According to the United Nations Industrial Development Organization, only three African countries have entered the “new industrialized economies”. Most African countries are still in the initial stage of industrialization, dominated by resource development, raw material processing, and manufacturing [50]. At this stage, the industrial economy and energy consumption grow synergistically. According to the International Energy Agency (IEA) and the *African Statistical Yearbook*, Africa’s total manufacturing output value increased from USD 115.3 billion in 2000 to USD 264 billion in 2018, an increase of 128.9%. At the same time, the total coal consumption by sector increased by 29.37%, from 19,092 Kote to 24,699 Kote. Particulate pollution from energy production, industrial operations, and car exhaust emissions brought significant environmental costs. Based on the method of the OECD National Environment Agency, Roy (2016) estimates that the total annual deaths caused by particulate pollution in Africa increased by 250,000 from 1990 to 2013 [51]. This information confirmed that most of Africa is still in the rising stages of “EKC”, and air quality negative effects, industrial construction, energy consumption coexist for a long time, which provides a powerful explanation for the significant impact of economic construction on PM_2.5_ concentrations in Africa [52].

The SEU results reflect the significant contributions of economic urbanization to PM_2.5_ concentrations of adjacent areas. Two reasons can explain such a finding. First, with the support of hydrodynamics, air pollutants produced by regional economic activities diffuse to neighboring areas due to wind forces [53]. Second, international trade has accelerated the globalization of emissions and pollution [8]. The value of goods exports and non-factor services increased by 105% from 2000 to 2018, reaching USD 560.8 billion. Rising economic contact and border trading activities between countries have contributed to increase the transmission of PM_2.5_ pollutants and pollutant components among neighboring countries [54]. This suggests that Africa’s PM_2.5_ pollution problem should not only focus on local urbanization but also include cross-regional strategies and policies to address air quality concerns.

The DEU of land urbanization has generally increased but is slightly lower than economic urbanization in 2018. This is most likely caused by two main reasons: construction dust pollution and green land shrinkage. On the one hand, rapid agglomeration of population and economic activities boost the construction of residential houses, industrial plants, and roads in cities. These urban constructions produce loads of road and construction dust, adversely impacting urban air quality [55]. On the other hand, increased urban building density and new land development accelerate green space fragmentation to a certain extent [56]. As a result, the functions of absorption and regulation of suspended particulate matter by vegetation is considerably reduced [57], and also indirectly affects the diffusion and sedimentation of particulate matter by increasing the heat island effect [58], which adversely impacts the region’s air quality.

Why did the interactions with multi-dimensional urbanization result in non-linear enhancement to PM_2.5_ concentrations? The PM_2.5_ pollution in Africa is not caused simply by the independent and direct impact of any single urbanization metric but rather the product of interactions of various urbanization factors. The systemicity and inherent synergy of urbanization may explain this phenomenon (Figure 10). The principal elements of urbanization, population, economy, and land serve as the basic support, source, and space carrier of urbanization, respectively [59,60]. The changes in residents’ lifestyles will accelerate the transformation of the socio-economic structure and urban space development and promote industrial growth and land use expansion. Among them, the socio-economic structure acts on atmospheric ecosystem by industrial pollution and energy consumption. Urban space development shows as increased urban land development and rising housing demands, which has negative effects on air quality in the form of green space shrinkage, building density growth, and construction dust [56]. Residents’ lifestyles often act on the air quality system in the form of household fuel and domestic waste combustion [61]. As a result, these various socio-economic activities influence and coordinate each other under human–land system science and ultimately affect the spatio-temporal evolution of regional PM_2.5_ concentrations. The significant interaction of multi-dimensional urbanization-affected PM_2.5_ concentrations also confirms the necessity and rationality to analyze the impact of urbanization on PM_2.5_ concentrations using multi-dimensional perspectives. Urban environmental planning and management should also integrate the population, economic, and land optimization measures to coordinate the relationship between urban development and air quality.

### 4.2. Policy Implications

The results showed that Africa’s urbanization had a significant positive impact on PM_2.5_ concentrations and has been increasing over the years. Various dimensions of urbanization exhibited interaction effect and spillover effects, which were generally stronger than the direct effects. How to promote the coordinated development of urbanization and air quality system in Africa? We believe that the more important thing for Africa’s air quality management is not only limited in understanding pollution mechanisms, but also addresses the negative environmental effects of urbanization under the condition of a weak economy. While traditional problems, such as diseases, poverty, hunger, and political turmoil, often limit many African countries to heavily invest in environmental protection and management, air quality concerns and other environmental problems should not be ignored. The urbanization path of treatment after pollution is obviously incompatible with the current resource and climate-constrained world order and the sustainability goals of urbanization. Africa has already paid a lot of economic costs and residents’ health for air pollution. According to a report (Roy, 2016), in 2013, the economic cost of air pollution in Africa was estimated at USD 447 billion, resulting in some 250,000 deaths [51]. As a result, air quality protection and environmental management must be promoted while stimulating economic growth and urban development [62]. Under our results and limited economic conditions, some works still remain to be proposed for promoting coordinated development of urbanization and air quality in Africa:
(1)Given the strong spillover effects across regions, integrated regional planning in air quality management must be strengthened among African countries. In particular, countries along the Gulf of Guinea and the Sahara Desert should enhance the joint action capacity for air pollution control, monitoring, and mitigation strategies. Developing emission inventories and environmental risk assessment systems within the framework of Agenda 2063 are also critical.(2)The interaction of multi-dimensional urbanization has a substantial amplifying effect on PM_2.5_ concentrations. As a result, air quality management in Africa should integrate various urban aspects, such as population lifestyle transformation, land structure optimization, and industrial upgrading. Policymakers should consider comprehensive urban development planning, domestic waste management, urban green space maintenance, and energy conservation, as well as the impact of cross-regional trade on PM_2.5_ transmission.(3)Urbanization complexity requires strong sectoral collaboration to effectively manage air pollution. Experiences in PM_2.5_ governance from other countries should be considered, and governments should promote pollution traceability and accountability. The coordinated management of pollution sources from different sectors should be further strengthened, including those in trade, greening, construction, industrial production, and transportation.


### 4.3. Research Limitations and Future Directions

There are several limitations in this study. First, although we explored the multi-dimensional urbanization effects on PM_2.5_ concentrations in Africa and discussed the reasons of the varying impacts of multi-dimensional urbanization, we did not reveal the threshold levels of population, land, and economic urbanization at which coordinated development of urbanization and air quality can be achieved. Coordinated response mechanisms between urbanization and air quality can be more effective in achieving sustainable development in Africa if they utilize the threshold response function and pollution warning model. Second, the analyses of the driving impact and interactions were based on the entire African continent. Causing the driving impact and spillover effects of multi-dimensional urbanization in particular, geographic areas are difficult to be analyzed. Extending the study area to a particular country or city in the future can help to reveal the interaction between urbanization and PM_2.5_ concentrations in small-scale areas and address local air pollution problems.

Despite these limitations, under the wave of global air pollution governance, this study introduced the driving analysis of urbanization on PM_2.5_ concentrations in Africa, and identified the different impacts of urbanization on the local and neighboring areas. For case with difficulty in data acquisition, *pd*, *isc*, and *ntl* were used to characterize the multiple dimensions of urbanization, providing a deeper understanding of the impact of multi-dimensional urbanization on PM_2.5_ concentrations in Africa. In addition, the empirical analysis of the different impacts and interaction of multi-dimensional urbanization on PM_2.5_ concentrations can be extended to other environmental issues, such as carbon emissions, carbon neutrality, and biodiversity conservation. The discussion on handling the relationship between urban development and PM_2.5_ pollution in underdeveloped areas can also be extended to other economically developing areas.

## 5. Conclusions

Previous studies have analyzed the impact of urbanization on PM_2.5_ concentrations in many areas. However, most of the research has overlooked the comprehensive impact of the various dimensions of urbanization (i.e., population, land, and economic) on PM_2.5_ concentrations, particularly in Africa. They seldom pay attention to the direct and spillover effects of multi-dimensional urbanization on PM_2.5_ concentrations, lacking the quantitative evaluation to the interaction effects of different dimensions of urbanization on PM_2.5_ concentrations. Therefore, this study explored the spatio-temporal evolution of PM_2.5_ concentrations and multi-dimensional urbanization at the grid and administrative levels and explored the driving mechanisms of PM_2.5_ concentrations. We also analyzed the varying direct and spillover impacts and interaction effects of multi-dimensional urbanization on PM_2.5_ concentrations. The results show that the areas exceeding the PM_2.5_ pollution standard (>35 μg/m^3^) increased over the study period 2000–2018 and can be found mainly along the Gulf of Guinea and the Sahara Desert. The influence of urbanization on PM_2.5_ concentrations has gradually increased and stayed dominant, and the economic urbanization level had the strongest impact on PM_2.5_ concentrations among the various urbanization dimensions. The results also show that the spillover effect of multi-dimensional urbanization was stronger than the direct effect, and Africa’s PM_2.5_ pollution problem was the product of interactions of various urbanization factors.

## Figures and Tables

**Figure 1 ijerph-18-09389-f001:**
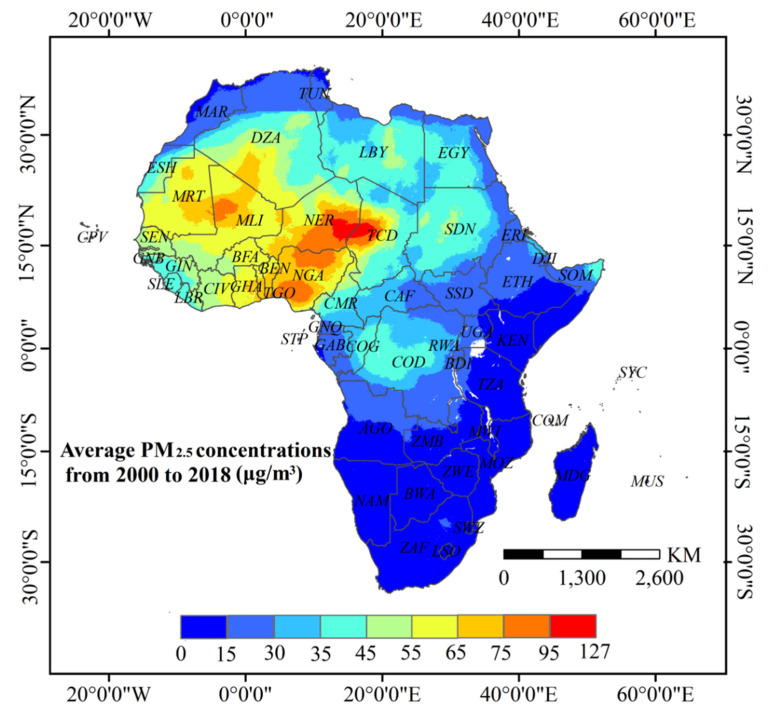
Spatial distribution of average PM_2.5_ concentrations in Africa from 2000 to 2018.

**Figure 2 ijerph-18-09389-f002:**
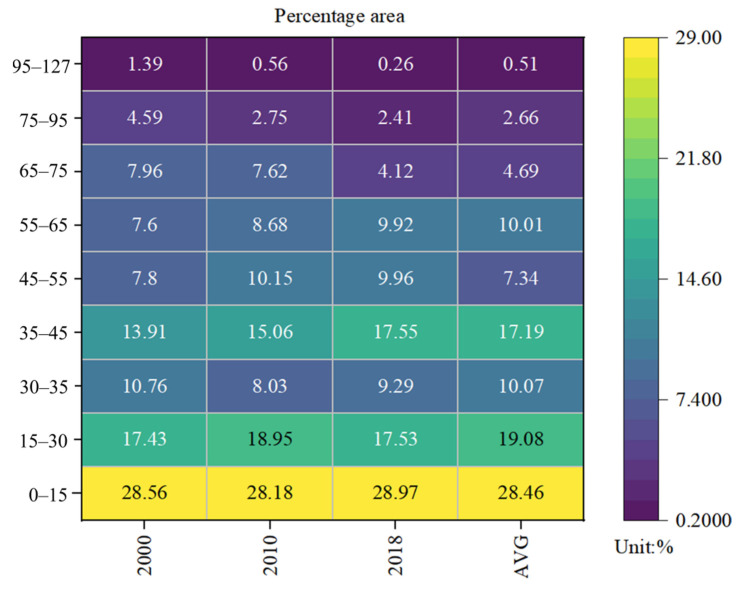
The percentage of the classified area of the annual average concentrations of PM_2.5_ in Africa from 2000 to 2018.

**Figure 3 ijerph-18-09389-f003:**
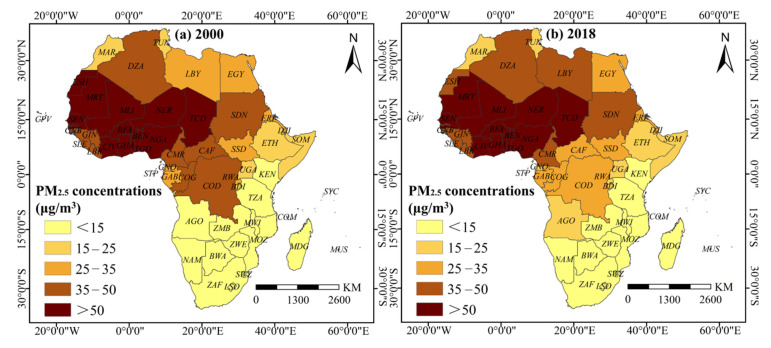
Spatial pattern of PM_2.5_ concentrations in African countries in 2000 and 2018.

**Figure 4 ijerph-18-09389-f004:**
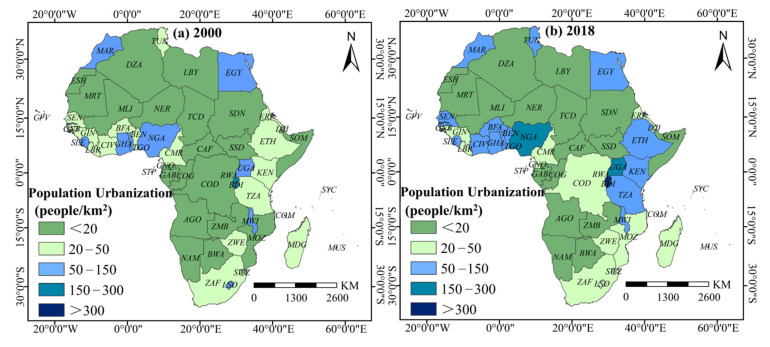
Spatial pattern of population urbanization level in African countries in 2000 and 2018.

**Figure 5 ijerph-18-09389-f005:**
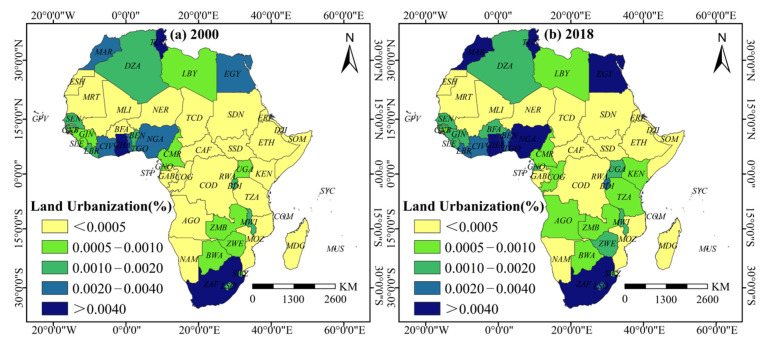
Spatial pattern of land urbanization level in African countries in 2000 and 2018.

**Figure 6 ijerph-18-09389-f006:**
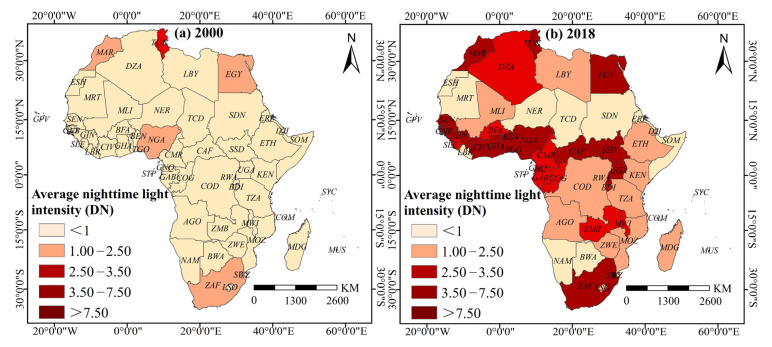
Spatial pattern of economic urbanization level in African countries in 2000 and 2018.

**Figure 7 ijerph-18-09389-f007:**
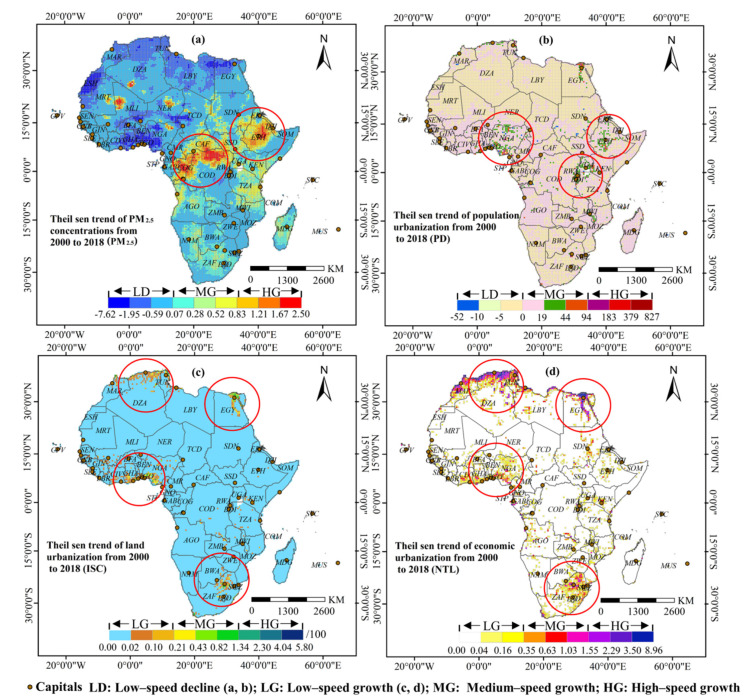
Spatial evolution trend of PM_2.5_ concentrations (**a**), population urbanization (**b**), land urbanization (**c**), economic urbanization (**d**) in African countries from 2000 to 2018.

**Figure 8 ijerph-18-09389-f008:**
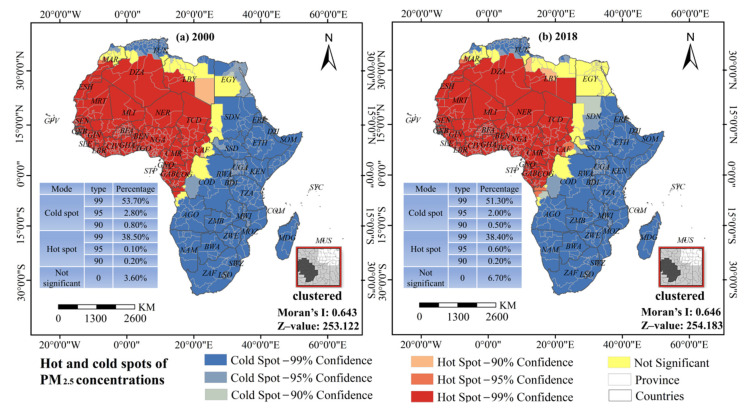
Hot and cold spots distributions of PM_2.5_ concentrations in Africa in 2000 and 2018.

**Figure 9 ijerph-18-09389-f009:**
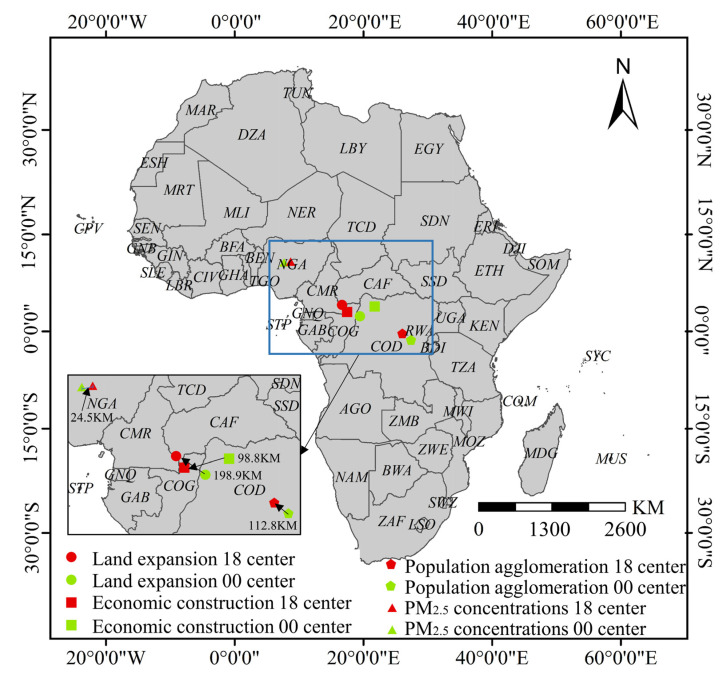
The evolution of the gravity center of PM_2.5_ concentrations and multi-dimensional urbanization in 2000 and 2018; for calculation formula, see Li et al. (2019) [48].

**Figure 10 ijerph-18-09389-f010:**
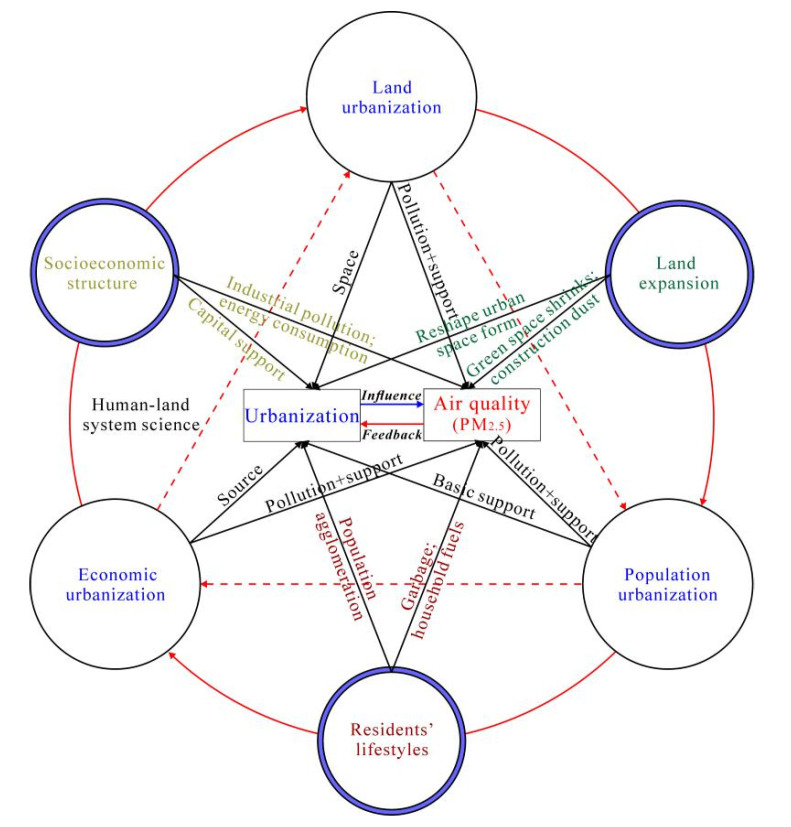
Influence mechanism of urbanization system on PM_2.5_ concentrations.

**Table 1 ijerph-18-09389-t001:** Variables setting and unit.

Variable Category	Variable	Abbreviation	Measurement Unit
Urbanization	Population density	*pd*	people/km^2^
	Impervious surface coverage	*isc*	%
	Nighttime light intensity	*ntl*	DN
Natural variable	PM_2.5_ concentrations	*PM* _2.5_	μg/m^3^
	Normalized differential vegetation index	*ndvi*	-
	Cumulative precipitation	*pre*	mm
	Elevation	*ele*	m
	Slope	*slope*	-

**Table 2 ijerph-18-09389-t002:** The judgment basis of interaction mode.

Interaction	Judgment Basis
Non-linear reduction	P(A∩B) < min(P(A),P(B))
Single-factor nonlinearity reduction	min(P(A),P(B)) < P(A∩B) < max(P(A),P(B))
Two-factor enhancement	P(A∩B) > max(P(A),P(B))
Independent	P(A∩B) = P(A) + P(B)
Non-linear enhancement	P(A∩B) > P(A) + P(B)

Notes: A and B are urbanization factors for any two different dimensions.

**Table 3 ijerph-18-09389-t003:** The list of countries and regions in Africa.

Regions	Countries and Regions
Northern Africa	Egypt, Libya, Tunisia, Algeria, Morocco, Sudan, Western Sahara
Eastern Africa	Eritrea, Ethiopia, South Sudan, Djibouti, Somalia, Kenya, Uganda, Rwanda, Burundi, Tanzania, Madagascar, Zambia, Zimbabwe, Malawi, Mozambique, Seychelles, Mauritius, Comoros
Western Africa	Nigeria, Benin, Ghana, Togo, Côte d’Ivoire, Liberia, Sierra Leone, Guinea, Guinea-Bissau, Senegal, Gambia, Mauritania, Mali, Niger, Cape Verde, Burkina Faso
Central Africa	Angola, Congo, Congo Dem.Republic, Equatorial Guinea, Gabon, Central African Republic, Chad, Cameroon, Sao Tome and Principe
Southern Africa	Botswana, Namibia, South Africa, Swaziland, Lesotho

**Table 4 ijerph-18-09389-t004:** Spatial regression model estimates of the driving factors of PM_2.5_ concentrations in 2000, 2010, and 2018.

Variables	SDM_2000	SDM_2010	SDM_2018
*lncmpu*	0.051 ***	0.054 ***	0.097 ***
*lnndvi*	−0.022 ***	−0.037 ***	−0.041 ***
*lnpre*	−0.017 ***	−0.007 *	0.015 **
*lnele*	−0.03 ***	−0.018 ***	−0.034 ***
*lnslope*	−0.038 ***	−0.062 ***	−0.060 ***
*W*cmpu*	0.068 ***	0.099 **	0.122 *
*W*ndvi*	0.014 ***	0.024 ***	0.020 ***
*W*pre*	0.042 ***	0.038 ***	0.031 ***
*W*ele*	0.012 ***	0.014 ***	0.028 ***
*W*slope*	−0.050 ***	−0.018 ***	−0.033 ***
R-squared	0.961 (SLM)	0.956 (SLM)	0.954 (SLM)
	0.961 (SEM)	0.956 (SEM)	0.955 (SEM)
	0.962 (SDM)	0.956 (SDM)	0.957 (SDM)
Log-L	24,236.9 (SLM)	24,428.8 (SLM)	24,343.7 (SLM)
	24,142.9 (SEM)	24,382.1 (SEM)	24,267.9 (SEM)
	24,273.9 (SDM)	24,467.1 (SDM)	24,384.3 (SDM)
LR-SLM	38,347.6 ***	34,852.9 ***	34,948.5 ***
LR-SEM	38,159.6 ***	34,759.6 ***	34,797.1 ***

Notes: cmpu indicates comprehensive urbanization index, which was obtained as follows: first, the mean standardized economic urbanization, land urbanization, and population urbanization level were calculated; and second, this sum value was standardized to obtain the final comprehensive urbanization index. *, **, and *** indicate the significance at the confidence level of 10%, 5%, and 1%, respectively.

**Table 5 ijerph-18-09389-t005:** Impact of multi-dimensional urbanization on PM_2.5_ concentrations in 2000, 2010, and 2018.

Row Number	Type of Urbanization	Variable	Indicator Abbreviation	Direct Effect (DEU)			Spillover Effect (SEU)		
				2000	2010	2018	2000	2010	2018
1	Population urbanization	Population density	*lnpd*	−0.005 (−0.410)	−0.001 (−0.147)	−0.002 (−0.224)	−0.251 (−0.399)	−0.783 (−1.511)	−0.265 (−0.512)
2	Land urbanization	Impervious surface coverage	*lnisc*	0.015 * (1.909)	0.030 *** (4.174)	0.026 *** (2.325)	−1.015 *** (−2.325)	0.406 (0.966)	−0.459 (−0.682)
3	Economic urbanization	Nighttime light intensity	*lnntl*	0.020 *** (4.225)	0.017 *** (4.007)	0.027 *** (5.671)	1.241 *** (3.103)	1.587 *** (3.684)	2.464 *** (4.228)

Notes: *t*-statistics in parentheses. * and *** indicate the significance at the confidence level of 10% and 1%, respectively.

**Table 6 ijerph-18-09389-t006:** The interaction effect of multi-dimensional urbanization factor on PM_2.5_ concentrations.

Year	P_1_ = A∩B	P_2_ = A + B	Comparison Result	Interaction Types
2000	PU∩LU = 0.07	P(0.003) + LU(0.029) = 0.032	P > PU + LU	Nonlinear enhancement
	PU∩EU = 0.12	PU(0.003) + EU(0.07) = 0.073	P > PU + EU	Nonlinear enhancement
	LU∩EU = 0.14	LU(0.029) + EU(0.07) = 0.099	P > LU + EU	Nonlinear enhancement
2010	PU∩LU = 0.11	PU(0.009) + LU(0.031) = 0.04	P > PU + LU	Nonlinear enhancement
	PU∩EU = 0.21	PU(0.009) + EU(0.08) = 0.017	P > PU + EU	Nonlinear enhancement
	LU∩EU = 0.11	LU(0.031) + EU(0.08) = 0.039	P > LU + EU	Nonlinear enhancement
2018	PU∩LU = 0.09	PU(0.005) + LU(0.017) = 0.022	P > PU + LU	Nonlinear enhancement
	PU∩EU = 0.63	PU(0.005) + EU(0.42) = 0.425	P > PU + EU	Nonlinear enhancement
	LU∩EU = 0.46	L(0.017) + EU(0.42) = 0.437	P > LU + EU	Nonlinear enhancement

Notes: PU, the *p*-value of population urbanization; LU, the *p*-value of land urbanization; EU, the *p*-value of economic urbanization. The confidence level of all variables exceeds 95%.

## Data Availability

The data used to support the findings of this study are available from the corresponding author upon request.

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
