# Peer review of "The Driving Influence of Multi-Dimensional Urbanization on PM2.5 Concentrations in Africa: New Evidence from Multi-Source Remote Sensing Data, 2000–2018"

_ijerph, 2021, doi:10.3390/ijerph18179389_

Round 1

Reviewer 1 Report

Some small corrections and remarks:

There is a good update of the references, there is even one from 2021, but none has the DOI link.

Line 150: where is “... population density , which was...” shoud be “... population density, which was... ”

Line 226: I think there is a mistake in this sentece. Where is “Values above zero indicate hot spots are while values below zero indicate cold spots.” maybe it shoud be “Values above zero indicate hot spots while values below zero indicate cold spots.”

Line 250: dtat  is not defined

Line 621: where is “... constrcution at which can achieve ...” shoud be “... construction at which can achieve ...”

Reference 14: date is not in bold

Reference 33: date is not in bold

Author Response

Some small corrections and remarks:

Point 1:There is a good update of the references, there is even one from 2021, but none has the DOI link.

Response 1: Thank you for your recognition of the references. According to your suggestion, we have added DOI links to all references, see the References section for details.

Point 2: Line 150: where is “... population density , which was...” shoud be “... population density, which was... ”

Response 2: Thank you for your kind reminder. We removed the space based on your suggestion and checked the sentence. See line148.

Point 3: Line 226: I think there is a mistake in this sentece. Where is “Values above zero indicate hot spots are while values below zero indicate cold spots.” maybe it shoud be “Values above zero indicate hot spots while values below zero indicate cold spots.”

Response 3: Yes, you are right. According to your suggestion, we modified this sentence to "Values above zero indicate hot spots while values below zero indicate cold spots". See line224.

Point 4: Line 250: dtat  is not defined

Response 4: Thanks for your suggestion. Our original intention is to illustrate the advantages of SDM in model suitability. According to your suggestion, we revised the content of this sentence and changed it to "If both are rejected, the SDM will be more suitable for the regression analysis of this study", see line 247.

Point 5: Line 621: where is “... constrcution at which can achieve ...” shoud be “... construction at which can achieve ...”

Response 5: Yes, here is a spelling error. We have corrected the spelling according to your suggestion, see 617.

Point 6: Reference 14: date is not in bold

Response 6: Thank you for your suggestion. We have revised the format according to your suggestion, see Reference14 for details.

Point 7: Reference 33: date is not in bold

Response 7: Thank you for your suggestion. We have revised the format according to your suggestion. See Reference33 for details.

Reviewer 2 Report

I have had difficulty in finding any novelty in the analysis of your data. It may, in part, be due to the looseness of your jargon. For example, the term "interaction of multi-dimensional human activities," is so broad as to be meaningless. You may want to change your phrasing to more specific terms such as traffic congestion, industrial output and electrical power production and in turn, focus on more specific causes that can better be individually identified and spefically addressed by politicans, planners and regulators. It seems to me that it is not possible from using the phrase "interaction of multi-dimensional human activities," that anyone would have any useful insight as to where to begin addressing the problem you have too broadly described. Whether you have access to such specific data is a concern that I have in offering this suggestion but without something more specific in your analysis the conclusions that can be drawn from the work you have presented are simply not novel enough to support publication.

Author Response

Point: I have had difficulty in finding any novelty in the analysis of your data. It may, in part, be due to the looseness of your jargon. For example, the term "interaction of multi-dimensional human activities," is so broad as to be meaningless. You may want to change your phrasing to more specific terms such as traffic congestion, industrial output and electrical power production and in turn, focus on more specific causes that can better be individually identified and spefically addressed by politicans, planners and regulators. It seems to me that it is not possible from using the phrase "interaction of multi-dimensional human activities," that anyone would have any useful insight as to where to begin addressing the problem you have too broadly described. Whether you have access to such specific data is a concern that I have in offering this suggestion but without something more specific in your analysis the conclusions that can be drawn from the work you have presented are simply not novel enough to support publication.

Response: Thank you for your pertinent suggestions.

(1) What needs to be clarified is that urbanization is the original theme of this research. At your suggestion, we changed the research theme from "multi-dimensional human activities" to "multi-dimensional urbanization" (population, land and economic urbanization) again. And the original title has also been adjusted to "The driving influence of multi-dimensional urbanization on PM2.5 concentrations in Africa: New evidence from multi-source remote sensing data, 2000-2018". Related pictures and research content have also been adjusted. It is worth noting that the concept of multi-dimensional urbanization has been mentioned in the study of Du et al., 2019, and the feasibility of reflecting population, land, and economic urbanization with population density, impervious surface coverage, and nighttime light intensity has been practiced and applied in previous studies (Peng et al., 2020; Zhang et al., 2021; Peng et al., 2017).

10.1016/j.jclepro.2019.02.222

10.1016/j.scitotenv.2017.06.218

10.1016/J.ECOSER.2021.101274

10.1016/j.ecoser.2020.101139

To be sure, we think you are right. Specific terms such as traffic congestion, industrial output and electrical power production may be more attractive and may be more advantageous in a practical sense. However, it is difficult to obtain remote sensing data related to Africa, and the population density, impervious surface coverage and nighttime light intensity we use are difficult to accurately reflect the microscopic development status of the area's traffic congestion and industrial output, etc. We have optimized the academic terms of the full article according to your suggestions, especially adjusted the analysis content with reference to the research of Du et al., 2019. It should be explained again that the highlight of this research is not the innovation of the analysis content, but in the context of difficulties in obtaining relevant data in Africa, for the first time the research target is aimed at the impact of multi-dimensional urbanization (population, land and economic urbanization) in Africa on air pollution. At the same time, we focused on the differentiated impacts and internal interactions of urbanization in the local and neighboring regions, which are rare in previous African studies, and provided support and new evidence for the need for African countries to carry out cross-regional joint treatment of air pollution. We sincerely thank you for your comments, sincerely hope that our explanations and modifications can meet your requirements, and sincerely hope that you can further recognize our research work.

Reviewer 3 Report

The manuscript is novel and will be of interest to the readers. It has a clear introduction and a well-defined objective. The methods are robust and the conclusions are supported by the results. It is a study that integrates several disciplines, as well as the use of software and big-data analysis on PM2.5. Few studies involve the analysis of an entire continent.

I suggest a general review to identify minimal editing errors:
Line 250. Data preprocessing?
Figure 2. Colors are confusing (blue and purple?). I suggest modifying so as not to confuse readers.
Figure 4. person / km2?

The manuscript is good. I am sure it will be a highly cited manuscript. Congratulations to all the authors!

Author Response

The manuscript is novel and will be of interest to the readers. It has a clear introduction and a well-defined objective. The methods are robust and the conclusions are supported by the results. It is a study that integrates several disciplines, as well as the use of software and big-data analysis on PM2.5. Few studies involve the analysis of an entire continent.

I suggest a general review to identify minimal editing errors:

Point 1: Line 250. Data preprocessing?

Response 1: Thank you for your suggestion, we reviewed the content. According to your suggestion, we corrected it to "The following procedures were undertaken to improve the rationality for spatial regression analysis", see line 249.

Point 2: Figure 2. Colors are confusing (blue and purple?). I suggest modifying so as not to confuse readers.

Response 2: Thank you for your suggestion. We agree with your point of view. According to your suggestion, we have modified the original figure. See Figure. 2 for details.

Point 3: Figure 4. person / km2?

Response 3: Thank you for your reminder. We reviewed the original content and changed the unit to people/km2, see Figure. 4. The use of this unit refers to the following references and the data description by the Resources and Environmental Science and Data Center of Chinese Academy of Sciences (https://www.resdc.cn/data.aspx?DATAID=251).

10.1016/j.scitotenv.2019.05.352

10.1016/j.ecoser.2020.101139

Point 4: The manuscript is good. I am sure it will be a highly cited manuscript. Congratulations to all the authors! 

Response 4: Thank you very much for your valuable suggestions and comments, which have played an important role in improving the quality of this research.

Reviewer 4 Report

This manuscript focuses on a case study that analyzes the spatial distribution of human activities and particulate pollution, as well as their statistical interrelationships, in Africa for 2010-18. The topic is important and relevant, and the paper is generally well-written and well-organized. However, there are a few methodological issues and weaknesses that render the statistical findings somewhat questionable and make this paper unsuitable for publication in its current form. My main concerns with the spatial/statistical analysis and related recommendations are listed below:

1-The measurement of spatial dependence and results of both spatial autocorrelation and spatial regression analysis are highly dependent on how spatial interaction (i.e., the weights matrix) is defined. It is mentioned (on page 5) that Fixed Distance and Euclidean distance methods were utilized, but no details or justification is provided. A more detailed discussion is needed to explain why these particular methods were chosen, how exactly they were implemented, whether other approaches (e.g., contiguity-based) were considered, and the assumptions/limitations of the distance-based methods. Without this information, it is difficult to evaluate the reliability and validity of the statistical findings.

2-The results of the LM test and robust LM test from the OLS estimates are used to justify the need for spatial regression models (page 12). A better and more valid approach is to test the OLS model residuals for spatial autocorrelation in order to make this determination. Many prior studies, for example, have used the statistical significance of the residual Moran’s I for this purpose. I recommend the authors report this measure (supported by an appropriate significance test) to demonstrate the need for regression models that account for spatial dependence.

3-On a related note, the spatial regression model results do not clearly demonstrate if spatial dependence or autocorrelation has been adequately accounted for in the spatial models presented in the paper. The authors need to include appropriate measures of spatial autocorrelation (e.g., Moran’s I of regression residuals) and stronger evidence to indicate that the statistically significant spatial autocorrelation observed in the OLS model was rendered non-significant in the spatial models.

4-I am not entirely satisfied with how the authors have assessed multicollinearity in their models. VIFs are not very effective in fully capturing the severity of multicollinearity in a multivariable model since it focuses on individual variables. Why not report the condition index instead? This will be based on the collection of all independent variables included in the model and will truly indicate whether or not there are serious problems of collinearity that warrant some attention in the analytical process. This multicollinearity condition index for each model should be reported in the text and Table 4.

Author Response

This manuscript focuses on a case study that analyzes the spatial distribution of human activities and particulate pollution, as well as their statistical interrelationships, in Africa for 2010-18. The topic is important and relevant, and the paper is generally well-written and well-organized. However, there are a few methodological issues and weaknesses that render the statistical findings somewhat questionable and make this paper unsuitable for publication in its current form. My main concerns with the spatial/statistical analysis and related recommendations are listed below:

Point 1:The measurement of spatial dependence and results of both spatial autocorrelation and spatial regression analysis are highly dependent on how spatial interaction (i.e., the weights matrix) is defined. It is mentioned (on page 5) that Fixed Distance and Euclidean distance methods were utilized, but no details or justification is provided. A more detailed discussion is needed to explain why these particular methods were chosen, how exactly they were implemented, whether other approaches (e.g., contiguity-based) were considered, and the assumptions/limitations of the distance-based methods. Without this information, it is difficult to evaluate the reliability and validity of the statistical findings.

Response 1: Thanks you for your valuable comments. As you said, the choice of spatial distance method is an important part of spatial autocorrelation analysis. By calculating the distance between samples, the spatial similarity of different samples can be estimated. Both spatial relations conceptualization and spatial weight construction were automatically generated in the spatial autocorrelation module of ArcGIS10.6 software. There are two reasons for choosing Fixed Distance for conceptualization of spatial relations: (1) according to previous studies by scholars, the diffusion of PM2.5 pollution sources in the area usually has a significant distance attenuation effect. Fixed Distance method was not only widely used by geography scholars, but also can include all the elements within the specified boundary of the element in the analysis, and the elements outside the critical distance will be excluded. This feature makes it applicable in this research; (2) by comparing the measurement results of different spatial conceptualization methods, it was found that the measurement results based on Fix Distance are more in line with the conclusions of Figure. 2 and Figure. 3 in terms of spatial expression.

The spatial weight matrix is also automatically implemented in the ArcGIS’s spatial autocorrelation module. The calculation of spatial weight matrix usually includes two distance methods, Euclidean and MANHATTAN, both of which can be used to describe the distance between two points. The reasons for choosing the Euclidean method are: (1) We re-examined the calculation results of the two different distance methods and found that the outputs of the two methods are not much different, and have less impact on the distribution pattern of PM2.5 cold and hot spots and Moran’I values in Africa. This may be due to the large range of the sample, which makes the difference between the calculation results based on the straight-line distance and the coordinate system distance relatively small. (2) We found that the Euclidean method is the default distance method in the ArcGIS’s spatial autocorrelation module, which was also mostly used by previous geography scholars for their spatial autocorrelation analysis. Thank you very much for your comments, which gave us a new understanding of the spatial autocorrelation model and strengthened our understanding of the conceptualization of spatial relations and the application of distance methods.

10.1016/j.jclepro.2018.03.198

10.1016/j.scitotenv.2019.05.352

Point 2: The results of the LM test and robust LM test from the OLS estimates are used to justify the need for spatial regression models (page 12). A better and more valid approach is to test the OLS model residuals for spatial autocorrelation in order to make this determination. Many prior studies, for example, have used the statistical significance of the residual Moran’s I for this purpose. I recommend the authors report this measure (supported by an appropriate significance test) to demonstrate the need for regression models that account for spatial dependence.

Response 2: thanks for your suggestions. The necessity of testing spatial regression models with LM and Robust LM was based on the research of Du et al., 2019. According to your suggestions, we further measured the residual Moran’ I of the OLS model, and the results showed that the residual Moran’ I of the model maintained a high level. This shows that the use of the classic linear regression model will ignore the impact of spatial autocorrelation, so it is necessary to use the spatial regression model for simulation calculations. The results see Line: 390-392.

Point 3: On a related note, the spatial regression model results do not clearly demonstrate if spatial dependence or autocorrelation has been adequately accounted for in the spatial models presented in the paper. The authors need to include appropriate measures of spatial autocorrelation (e.g., Moran’s I of regression residuals) and stronger evidence to indicate that the statistically significant spatial autocorrelation observed in the OLS model was rendered non-significant in the spatial models.

Response 3: Thanks for your suggestion. Based on your suggestion, we further verified the residuals Moran’I after passing the LM test to detect the autocorrelation of residuals and determine whether a spatial measurement model should be constructed to analyze the driving mechanism of PM2.5 concentrations. The result shows that the use of classical linear regression model to simulate the driving mechanism of PM2.5 concentrations usually ignores the influence of spatial dependence, which leads to deviations in the model simulation effect, so it is necessary to apply the spatial regression model to simulate and calculate.

Point 4: I am not entirely satisfied with how the authors have assessed multicollinearity in their models. VIFs are not very effective in fully capturing the severity of multicollinearity in a multivariable model since it focuses on individual variables. Why not report the condition index instead? This will be based on the collection of all independent variables included in the model and will truly indicate whether or not there are serious problems of collinearity that warrant some attention in the analytical process. This multicollinearity condition index for each model should be reported in the text and Table 4.

Response 4: Thanks for your suggestions. You are right that both VIF and Condition Index are important metrics to test the multiple collinearity of the model. We have considered the Condition Index in the calculation but did not report. According to your suggestion, we make a supplement in the article, see line: 248-251. We found that Condition Index for all variables were significantly less than 30, all passed the collinearity test. The following table shows the Condition Index of the regression variables in 2000, 2010 and 2018.

Condition Index of Regression Variables: 2000, 2010 and 2018

2000

2010

2018

Y

1

1

1

X1

2.048

2.048

2.304

X2

2.704

2.707

2.797

X3

3.317

3.312

3.407

X4

5.569

5.621

5.787

X5

9.974

9.621

9.247

Round 2

Reviewer 2 Report

I would agree that any information on Africa is an important contribution to the literature since so little has been done previously. That fact was hidden (and in some respects remains underemphasized) in the first version and so the most important aspect of the article was unclear. The current version is an improvement and so is acceptable. However, the authors could have still emphasized the issue more and rearranged the paper to more fully draw attention to this issue. It is unfortunate that there was not as strong a statement as the authors made in their comments back to me. It is a missed opportunity.

Author Response

Response: We sincerely thank you for your recognition and support for our revision work. With your help, we have achieved a lot of improvement in the quality of the article, and fully realize the importance of research expression. Based on your suggestions, we further elaborate on the significance of this research in the article, in order to draw readers’ attention to the fact that urbanization in Africa is aggravating air pollution on the local and neighboring regions. For details, see line: 18-20; 110-112; 117-118; 630-635.

Reviewer 4 Report

The authors have attempted to address the four key concerns identified in my previous review of the paper. While I am generally satisfied with the revisions and explanations, there are two issues that still need attention:

1-I agree with the explanation provided for my first comment (Point 1) in the Author Response section. But I would like to see some of this explanation and appropriate justification for using of Fixed Distance and Euclidean distance methods (lines 223-225 on page 5) included in the paper, as recommended previously.

2-I had also recommended including the Moran’s I of regression residuals as evidence to indicate that the statistically significant spatial autocorrelation observed in the OLS model was rendered non-significant in the spatial models (see Point 3). While the Moran's I has been reported only for the OLS model (lines 389-391), these values for the spatial models were not included. It is thus unclear if the residual Moran's I has become sufficiently small or non-significant in the spatial regression models.

Author Response

The authors have attempted to address the four key concerns identified in my previous review of the paper. While I am generally satisfied with the revisions and explanations, there are two issues that still need attention:

Point 1: I agree with the explanation provided for my first comment (Point 1) in the Author Response section. But I would like to see some of this explanation and appropriate justification for using of Fixed Distance and Euclidean distance methods (lines 223-225 on page 5) included in the paper, as recommended previously.

Response: Thank you for your suggestions. All your suggestions are very helpful to improve the quality of the manuscript. We once again express our sincere thanks to you. Based on your suggestion, we have strengthened the description of the reason for choosing the distance method in the article. For details, see line: 223-226.

Point 2: I had also recommended including the Moran’s I of regression residuals as evidence to indicate that the statistically significant spatial autocorrelation observed in the OLS model was rendered non-significant in the spatial models (see Point 3). While the Moran's I has been reported only for the OLS model (lines 389-391), these values for the spatial models were not included. It is thus unclear if the residual Moran's I has become sufficiently small or non-significant in the spatial regression models.

Response: Thank you for your suggestion. Based on your suggestion, we have added a description of the spatial model Moran’s I to show the necessity of the spatial regression model to replace the OLS model. For details, see line: 392-395.
